# MULTI-SEGMENTAL INFORMATIONAL CODING FOR SELF-SUPERVISED REPRESENTATION LEARNING

## ABSTRACT

Self-supervised representation learning aims to map high-dimensional data into a compact embedding space, where samples with similar semantics are close to each other. Currently, most representation learning methods maximize the cosine similarity or minimize the distance between different views from the same sample in an $\ell^2$ normalized embedding space, and reduce the feature redundancy via a linear correlation constraint. In this study, we propose MUlti-Segmental Informational Coding (MUSIC) as a new embedding scheme for self-supervised representation learning. MUSIC divides an embedding vector into multiple segments to represent different types of attributes, and each segment automatically learns a set of discrete and complementary attributes. MUSIC enables the estimation of the probability distribution over discrete attributes and thus the learning process can be directly guided by information measurements, reducing the feature redundancy beyond the linear correlation. Our theoretical analysis guarantees that MUSIC learns transform-invariant, non-trivial, diverse, and discriminative features. MUSIC does not require a special asymmetry design, a very high dimension of embedding features, or a deep projection head, making the training framework flexible and efficient. Extensive experiments demonstrate the superiority of MUSIC.

## 1  INTRODUCTION

Self-supervised representation learning (SSRL) is now recognized as a core task in machine learning with rapid progress over the past years (Bengio et al., 2013; LeCun et al., 2015). Deep neural networks pre-trained on large-scale unlabeled datasets via SSRL have demonstrated impressive characteristics, such as strong robustness (Hendrycks et al., 2019; Liu et al., 2021) and generalizability (Mohseni et al., 2020), and improving various downstream tasks. Among various pretext tasks, an effective approach is to drive semantically similar samples (*i.e.*, different transformations of the same instance) close to each other in the embedding space (Dosovitskiy et al., 2014; Wu et al., 2018; Tian et al., 2020b; Ye et al., 2019; Dwibedi et al., 2021). Simply maximizing the similarity or minimizing the Euclidean distance between embedding features of semantically similar samples tends to produce trivial solutions; *e.g.*, all samples have the same embedding features. Recently, various excellent methods have been proposed to learn meaningful representations feature and avoid trivial solutions. Contrastive learning (Hadsell et al., 2006; Oord et al., 2018) based methods, such as SimCLR (Chen et al., 2020a;b) and MoCo (He et al., 2020), have achieved great success by additionally minimizing the similarity between embeddings of the reference and negative samples, which requires either relatively large batches or a large memory bank (Wu et al., 2018; Misra & Maaten, 2020) for negative samples. To avoid using negative samples, BYOL (Grill et al., 2020) and SimSiam (Chen & He, 2021) developed clever techniques, such as the asymmetry network architecture, stop gradients, and momentum weight updating. Subsequent theoretical analysis (Wang & Isola, 2020; Zhang et al., 2021a; Richemond et al., 2020; Tian et al., 2021) have demonstrated why these techniques avoid trivial solutions and learn meaningful representations from different aspects. Clustering-based methods DeepCluster (Caron et al., 2018), SELA (Asano et al., 2019), SwAV (Caron et al., 2020) alternatively compute the cluster assignment of one view and optimize the network to predict the same assignment for other views of the same sample, where trivial solutions can be avoided via the even assignment of samples over different clusters. In another direction, W-MSE (Ermolov et al., 2021) and Barlow Twins (Zbontar et al., 2021) propose to drive self- or cross-correlation matrices towards the identity matrix, reducing the feature redundancy and learning

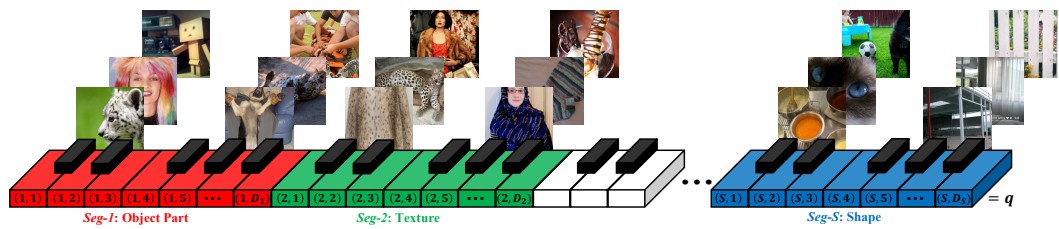

Figure 1: Illustration of the MUSIC vector with piano keys. The MUSIC embedding vector consists of multiple segments ($Seg$-1, ..., $Seg$-S) representing different types of attributes shown in different colors. Each segment is associated with a set of discrete attributes; *e.g.*, Seg-2 represents the texture attribute, and different units in Seg-2 specify different textural patterns, like dots, stripes, etc. Each segment is discretized with a one-hot vector $q(s,:)$.

meaningful features without requiring the asymmetry design. Most recently, VICReg (Bardes et al., 2022) constructs a novel loss function with three terms, i.e., invariance, variance, and covariance constraints that explicitly suppress trivial solutions. A theoretical study (Shwartz-Ziv et al., 2022) has given a deep analysis on why VICReg works well.

Fundamentally different from the current SSRL methods that normalize embedding features onto the unit hypersphere via $\ell^2$ norm and use cosine similarity as the metric, we propose MUlti-Segmental Informational Coding (MUSIC) for representation learning in a novel way. The motivation is based on the observation that an object can be represented by a set of attributes (Russakovsky & Fei-Fei, 2010). As illustrated in Fig. 1, we construct an embedding vector with multiple segments to represent different types of attributes; *e.g.*, $Seg$-1, $Seg$-2, and $Seg$-S represent object configuration, texture, and shape, respectively, and each segment instantiates a set of specific attributes; *e.g.*, $Seg$-2 represents samples with different textural patterns (dots, stripes, etc.). In other words, we discretize the feature variable by a segment of the one-hot vector implemented with softmax function, and the whole embedding vector consists of multiple such one-hot vectors. By doing so, MUSIC makes it possible to estimate the probability distribution over discrete units of each segment so that the information measures defined on probability distributions can be directly computed for both optimization and theoretical analysis. Two general properties are desired behind the illustration in Fig. 1: 1) samples can be classified into a set of different and discrete attributes in each segment; and 2) different segments discriminate samples using different classification criteria, which means that the mutual information between different segments is minimized, or equivalently the information/entropy of embedding features is maximized. To automatically learn such MUSIC embeddings for SSRL, we propose an entropy-based loss function based on the empirical joint probability distribution. Our information-theoretic analysis reveals why such meaningful features can be promoted while trivial solutions are avoided, which are consistent to the qualitative results in Appendix C.

The contributions of MUSIC are as follows. (1) MUSIC presents a new embedding scheme and enables a new information-theoretic optimization framework for SSRL. (2) Theoretical analysis ensures that the MUSIC embeddings are optimized to be transform-invariant, non-trivial, diverse, and discriminative. Importantly, MUSIC can minimize any form of dependency between feature variables beyond the linear correlation in current methods (Zbontar et al., 2021; Bardes et al., 2022). (3) Similar to Barlow Twins and VICReg, MUSIC does not require an asymmetry network architecture, negative samples in a large batch or a memory bank, gradient stopping, or momentum updating. (4) MUSIC does not need a very high dimension of embedding features or a deep projection head, significantly reducing the computational cost. (5) Extensive experimental results demonstrate the superiority of MUSIC on representative datasets in various evaluation settings.

## 2 METHODOLOGY

### 2.1 SELF-SUPERVISED LEARNING FRAMEWORK

Similar to W-MSE and Barlow Twins, in this study we adopt a twin architecture to learn the embedding features, where the same network is shared between two branches, as shown in Fig. 2. During training, input images $\mathbb{X} = \{x_i\}_{i=1}^N$ are mapped to two distorted sets $\mathbb{X}' = \{x_i'\}_{i=1}^N$ and

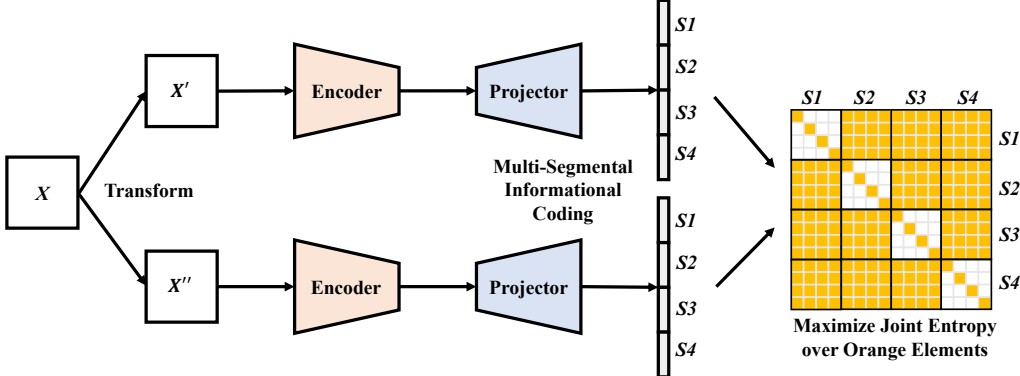

Figure 2: SSRL framework through multi-segmental informational coding optimized with maximum entropy. Here there are four segments and each segment consists of four units for illustration.

$\mathbb{X}'' = \{\boldsymbol{x}_i''\}_{i=1}^N$, where $N$ is the batch size. A common transformation distribution, which covers random crops combined with color distortions, the same as that in (Bardes et al., 2022), is used to generate training samples. Then, the two sets of distorted images $\mathbb{X}'$ and $\mathbb{X}''$ are respectively fed to two branches, each of which consists of an encoder $f(\cdot; \boldsymbol{\theta}_f)$ and a projector $g(\cdot; \boldsymbol{\theta}_g)$, where $\boldsymbol{\theta}_f$ and $\boldsymbol{\theta}_g$ respectively denote the parameters of the encoder and projector to be optimized. The outputs of the encoder are commonly used as the representation features. The projection head maps the representation features into the embedding space. Note that the presented method is not limited to this twin architecture, which can be extended to the two branches with different parameters, heterogeneous networks, or even different input modalities (e.g., text, audio, etc.) as studied in VICReg.

## 2.2 MULTI-SEGMENTAL INFORMATIONAL CODING

MUlti-Segmental Informational Coding (MUSIC) is a novel embedding scheme for SSRL. The embedding features of two transformed images are $\boldsymbol{z}_i' = g(f(\boldsymbol{x}_i'; \boldsymbol{\theta}_f); \boldsymbol{\theta}_g) \in \mathbb{R}^D$, and $\boldsymbol{z}_i'' = g(f(\boldsymbol{x}_i''; \boldsymbol{\theta}_f); \boldsymbol{\theta}_g) \in \mathbb{R}^D$ respectively, where $D$ is the feature dimension. Most of the existing SSRL methods normalize embedding features in the $\ell^2$ norm and then maximize their cosine similarity between the two transformed versions. Motivated by the observation described in Fig. 1, we divide the embedding feature $\boldsymbol{z}_i$ into multiple segments to represent different types of attributes, denoted by $\boldsymbol{z}_i(s, d), s = 1, \cdots, S, d = 1, \cdots, D_s$, where $S$ is the number of segments, $D_s$ is the dimension of the $s^{th}$ segment. In this study, we evenly split the embedding vector, i.e., $\forall s, D_s = D_S$, and the dimension of the whole embedding space is $D = D_S \times S$. In principle, uneven partitions can be applied as well given prior knowledge. To make attributes discrete and complementary, one-hot encoding is applied to each segment. Specifically, each segment is normalized to a score vector using the softmax function:

$$\boldsymbol{q}_i'(s', d') = \frac{\exp(\boldsymbol{z}_i'(s', d'))}{\sum_{d=1}^{D_S} \exp(\boldsymbol{z}_i'(s', d))}, \tag{1}$$

where $\boldsymbol{q}_i'(s', d')$ denotes the score of the image $\boldsymbol{x}_i'$ belongs to the $d'$-$th$ instantiated attribute in the $s'$-$th$ segment. The score vector $\boldsymbol{q}_i''(s'', :)$ for the other branch is computed in the same way. Thus, the MUSIC scheme can be interpreted as a combination of multiple classifiers or cluster operators that implement different classification criteria learned in a data-driven fashion.

## 2.3 ENTROPY LOSS

Since each type of attribute has a finite number of discrete instantiations, it is possible to estimate the probability distribution of a set of samples over each and every segment. Specifically, we can compute the empirical joint distribution $\boldsymbol{P}(s', s'', d', d'')$ between every two instantiated attributes within and across segments based on a set of samples as follows:

$$\boldsymbol{P}(s', s'', d', d'') = \frac{1}{N} \sum_{i=1}^N \boldsymbol{q}_i'(s', d') \boldsymbol{q}_i''(s'', d''), \tag{2}$$

where $\boldsymbol{P}(s', s'', d', d'')$ is computed as the statistical frequency of the sample having both the attribute-$d'$ in the segment-$s'$ and the attribute-$d''$ in the segment-$s''$ over $N$ samples. With the empirical joint probability distribution, information-theoretic metrics can be directly computed. Here two versions of the loss function are defined. The first version $L_{ent}$ is a pure joint entropy loss:

$$L_{ent} = \frac{1}{S^2} \sum_{s'=1}^{S} \sum_{s''=1}^{S} \sum_{d'=1}^{D_S} \sum_{d''=1}^{D_S} (1 - \mathbf{1}_{s'=s'', d'\neq d''}) \boldsymbol{P}(s', s'', d', d'') \log(\boldsymbol{P}(s', s'', d', d'')), \quad (3)$$

where $\mathbf{1}_{s'=s'', d'\neq d''}$ is an indicator function that equals to 1 if $s' = s''$ and $d' \neq d''$; otherwise, it is equal to 0. The empirical joint distribution can be denoted by a block matrix as shown in Fig. 2, where $(1 - \mathbf{1}_{s'=s'', d'\neq d''})$ means keeping the diagonal elements of the diagonal blocks and all elements of the off-diagonal blocks, as indicated by the orange area. Therefore, minimizing this loss function is maximizing the joint entropy over the selected elements. The following subsection will demonstrate the properties of the embedding features learned by optimizing this loss function.

To enhance the transformation invariance of features, we introduce an additional term to maximize the inner product between the embedding features from two transformations. Then, the second version of the loss function is defined as

$$L = L_{ent} - \lambda \frac{1}{NS} \sum_{i=1}^{N} \sum_{s=1}^{S} \sum_{d=1}^{D_S} \log(\boldsymbol{q}_i'(s, d) \boldsymbol{q}_i''(s, d)), \quad (4)$$

where $\lambda$ is a balancing factor. By default, we set $\lambda = 1$, which is in principle neither too small nor too large for a good balance. Minimizing the transformation invariance loss enforces the embedding features between two transformations of the same image to be consistent and encourages the embedding feature within each segment to be one-hot. Clearly, this additional term promotes transformation invariance and confident assignments over different attributes. Different from the statistical entropy measurement, this transformation invariance term imposes a sample-specific constraint. The transformation invariance can be also achieved by minimizing the cross-entropy between the two embedding vectors, *i.e.*, $-\frac{1}{NS} \sum_{i=1}^{N} \sum_{s=1}^{S} \sum_{d=1}^{D_S} \boldsymbol{q}_i'(s, d) \log(\boldsymbol{q}_i''(s, d))$. However, by using the cross-entropy the performance would be degraded, as reported in Subsection 4.2.

Our proposed method can be easily implemented, with a PyTorch-style pseudo-code in Appendix A. Next, let us theoretically analyze why the entropy loss optimizes informational embedding features as illustrated in Fig. 1.

## 2.4 ANALYSIS

The entropy loss function consists of two parts, including the entropy over diagonal elements of diagonal blocks and the entropy over all elements of off-diagonal blocks as illustrated by the orange area in Fig. 2, and can be formally expressed as

$$L_{ent} = \frac{1}{S} \sum_{s', s'', s'=s''} \sum_{d', d'', d'=d''} \boldsymbol{P}(s', s'', d', d'') \log(\boldsymbol{P}(s', s'', d', d'')) \\ + \frac{1}{S(S-1)} \sum_{s', s'', s'\neq s''} \sum_{d', d''} \boldsymbol{P}(s', s'', d', d'') \log(\boldsymbol{P}(s', s'', d', d'')). \quad (5)$$

For the first part, it can be demonstrated that its optimal solution is that $\forall i, s, d, \boldsymbol{q}_i'(s, d) = \boldsymbol{q}_i''(s, d)$, $\boldsymbol{q}_i'(s, :)$ and $\boldsymbol{q}_i''(s, :)$ are one-hot vectors, the statistical probability of the $s^{th}$ attribute type taking the $d^{th}$ instantiation is $\boldsymbol{p}(s, d) = \frac{1}{N} \sum_{i=1}^{N} \boldsymbol{q}_i(s, d) = \frac{1}{D_S}$, and $\boldsymbol{P}(s, s, d, d) = \frac{1}{D_S}$. The proof can be found in Appendix B. For the second part, it is intuitive that the optimal solution to maximize the joint entropy over the off-diagonal block items is $\forall s', s'', d', d'', s' \neq s'', \boldsymbol{P}(s', s'', d', d'') = \frac{1}{(D_S)^2}$; *i.e.*, a batch of samples are evenly assigned over each off-diagonal block.

**Transformation invariance**: The solution that $\forall i, s, \boldsymbol{q}_i'(s, :) = \boldsymbol{q}_i''(s, :)$ are one-hot vectors means that the learned MUSIC embeddings are invariant to transformations, and a sample tends to be confidently represented by a single instantiated attribute within each and every segment.

**Non-trivial solution**: The solution that $\frac{1}{N} \sum_{i=1}^{N} \boldsymbol{q}_i(s, d) = \frac{1}{D_S}$ means that a batch of samples are evenly assigned over different attributes in each segment as $\boldsymbol{q}_i'(s, :)$ and $\boldsymbol{q}_i''(s, :)$ are one-hot vectors.

Thus, the trivial solution that all samples have the same embedding features can be avoided. The discriminative encoding analyzed below also ensures MUSIC emebddings are non-trivial.

**Minimum redundancy**: As described in Fig. 1, different segments of the MUSIC embedding vector are expected to focus on diverse and complementary attributes. In other words, the redundancy or mutual information between any two segments should be minimized, which is a popular measure for feature selection (Peng et al., 2005). Specifically, it can be demonstrated that the redundancy or mutual information between any two segments is minimized when the optimal solution is obtained. Specifically, the mutual information $I(s', s'')$ between any two segments $s'$ and $s''$ is

$$
\begin{aligned}
I(s', s'') =& H(s') + H(s'') - H(s', s'') \\
=& - \sum_{d'=1}^{D_S} \boldsymbol{p}'(s', d') \log(\boldsymbol{p}'(s', d')) - \sum_{d''=1}^{D_S} \boldsymbol{p}''(s'', d'') \log(\boldsymbol{p}''(s'', d'')) \\
& + \sum_{d'=1}^{D_S} \sum_{d''=1}^{D_S} \boldsymbol{P}(s', s'', d', d'') log(\boldsymbol{P}(s', s'', d', d'')) \\
=& - log \frac{1}{D_S} - \log \frac{1}{D_S} + \log \frac{1}{(D_S)^2} = 0.
\end{aligned}
\tag{6}
$$

Thus, the mutual information is minimized and diverse attributes are learned. From another perspective, $\forall s', s'', d', d'', s' \neq s''$, we have $\boldsymbol{P}(s', s'', d', d'') = \frac{1}{(D_S)^2} = \boldsymbol{p}(s', d')\boldsymbol{p}(s'', d'')$, and thus the variables $\boldsymbol{q}(s', d')$ and $\boldsymbol{q}(s'', d'')$ are independent. The redundancy constraint was studied for W-MSE, Barlow Twins, and VICReg by minimizing the covariance or linear correlation. In contrast, our entropy-based loss function reduces the redundancy or mutual information in a non-linear way. Moreover, it can be derived that the optimal MUSIC embedding features have zero covariance between any two features in different segments and negative covariance between the features within the same segment; for details, please see Appendix B.

**Discriminative encoding**: Contrastive learning or instance discrimination has proven very effective for representation learning by maximizing the similarity between different transformations of the same instance while discriminating the reference from other instances. It is underlined that MUSIC is consistent to contrastive learning and discriminating instances in a novel way. Specifically, the optimal MUSIC embedding can totally encode $(D_S)^S$ different samples. In our default settings $D_S = 80, S = 102$ (See Section 3 for details), MUSIC can represent $80^{102}$ different samples. The maximized the joint entropy means that any two units from every two segments have the equal possibility to co-occur, that is, a batch of samples are evenly assigned to all possible embeddings. Since the number of all possible embeddings is much larger (2,048 vs $80^{102}$) than the batch size, it will be enforced to encode different instances with different embeddings. Like contrastive learning, it ensures non-trivial solutions. The difference lies in that contrastive learning differentiates instances by pushing the reference away from its negative instances, while MUSIC intrinsically assigns instances with different attribute codes in an information-principled manner.

In Appendix C, the individual MUSIC embeddings and the empirical joint probability matrix learned on the ImageNet dataset are visualized, showing that the empirical results are consistent with the above theoretical analysis. In summary, the MUSIC embedding features optimized with the entropy-based loss are transform-invariant, non-trivial, diverse, and discriminative.

## 3 IMPLEMENTATION DETAILS

For a fair comparison, we closely followed the implementation settings in VICReg to train MUSIC models. Specifically, the standard ResNet-50 backbone (He et al., 2016) was used as the encoder that outputs a representation vector of 2,048 units in the same training settings, including the data augmentation (random cropping, horizontal flip, color jittering, grayscale, Gaussian blur, solarization, with the same parameters in VICReg), the optimizer of LARS (You et al., 2017; Goyal et al., 2017) with a weight decay of $10^{-6}$ and the learning rate of $lr = batch\_size/256 \times base\_lr$, and the cosine decay schedule (Loshchilov & Hutter, 2016) from 0 with 10 warmup epochs towards the final value of 0.002. The base learning rate $base\_lr$ was set to 0.6 in our study. By default, we used a two-layer MLP projector (8,192-8,160), the number of segments $S = 102$, the segment dimension

$D_S = 80$, and $D = D_S \times S = 8,160$ (similar to the feature dimension of $8,192$ used by VICReg and Barlow Twins). The results were respectively analyzed for different feature dimensions, depths of projectors, batch sizes, segment dimensions, and numbers of training epochs. The effect of the single extra hyperparameter $D_S$ of MUSIC was evaluated as well. The SSRL models were trained on the 1,000-classes ImageNet dataset without labels and evaluated in various downstream tasks. All implementation details for downstream tasks are in Appendix G. Code will be made available.

## 4 RESULTS

### 4.1 EVALUATION RESULTS IN DIFFERENT TASKS

Table 1: **Comparison of SSRL methods on ImageNet for linear and semi-supervised classification**. Top-1 and Top-5 accuracies (in %) are reported. The best three results are underlined.

| Methods | Linear Classification | | Semi-supervised Learning | | | |
| --- | --- | --- | --- | --- | --- | --- |
| | Top-1 | Top-5 | Top-1 | | Top-5 | |
| | | | 1% | 10% | 1% | 10% |
| Supervised | 76.5 | - | 25.4 | 56.4 | 48.4 | 80.4 |
| MoCo (He et al., 2020) | 60.6 | - | - | 57.2 | 83.8 | |
| PIRL (Misra & Maaten, 2020) | 63.6 | - | - | - | - | - |
| CPC v2 (Henaff, 2020) | 63.8 | - | - | - | - | - |
| CMC (Tian et al., 2020a) | 66.2 | - | - | - | - | - |
| SimCLR (Chen et al., 2020a) | 69.3 | 89.0 | 48.3 | 65.6 | 75.5 | 87.8 |
| MoCo v2 (Chen et al., 2020c) | 71.1 | 90.1 | - | - | - | - |
| SimSiam (Chen & He, 2021) | 71.3 | - | - | - | - | - |
| SwAV (Caron et al., 2020) | 71.8 | - | - | - | - | - |
| InfoMin Aug (Tian et al., 2020b) | 73.0 | 91.1 | - | - | - | - |
| BYOL (Grill et al., 2020) | 74.3 | 91.6 | 53.2 | 68.8 | 78.4 | 89.0 |
| Barlow Twins (Zbontar et al., 2021) | 73.2 | 91.0 | 55.0 | 69.7 | 79.2 | 89.3 |
| VICReg (Bardes et al., 2022) | 73.2 | 91.1 | 54.8 | 69.5 | 79.4 | 89.5 |
| MUSIC (Ours) | 73.6 | 91.4 | 54.0 | 69.1 | 78.9 | 89.1 |

**Linear Classification on ImageNet.** Linear probing is the commonly used evaluation protocol that trains a linear classifier on top of the frozen representations to evaluate the performance of SSRL methods. Being consistent with Barlow Twins and VICReg, a ResNet-50 backbone was trained with the batch size of 2,048 for 1,000 epochs on the training set of ImageNet, and the linear classification results including Top-1 and Top-5 accuracies of different methods on the evaluation set are reported in Table 1. The difference from Barlow Twins and VICReg is that MUSIC used a two-layer MLP projector (8,192-8,160) instead of three layers (8,192-8,192-8,192). The performance of MUSIC is on par with the state of the art method BYOL that uses asymmetric techniques, such as an additional predictor and a momentum encoder. Note that the results of some excellent methods (Caron et al., 2020; Gidaris et al., 2021) based on multi-crop/multi-positive techniques are not included in Table 1. These techniques can usually boost performance further. The comparative results show that MUSIC achieves better results than Barlow Twins and VICReg, where all these three methods trained a twin architecture without using negative pairs or any asymmetric techniques.

**Semi-Supervised Classification on ImageNet.** We also evaluated MUSIC in the semi-supervised learning setting, where the pre-trained ResNet-50 with MUSIC was fine-tuned on subsets of ImageNet, including 1% and 10% of the full ImageNet dataset respectively, and all reported methods used the same subset images. Currently, MUSIC is not as good as Barlow Twins and VICReg in the semi-supervised learning settings, while it is better than BYOL and other compared methods, and significantly better than purely supervised learning without SSRL pretraining.

**Transfer Learning.** Transfer learning is another popular way for the evaluation of SSRL methods, including object detection, instance segmentation, and linear classification. Our results are reported in Table 2. It is noted that various studies have different setups for the object detection and instance segmentation tasks. Here we closely followed (Zbontar et al., 2021) selecting the same compari-

Table 2: **Transfer Learning**. For object detection and instance segmentation tasks, SSRL models pre-trained on ImageNet were used to initialize the backbone of the object detection and instance segmentation models on COCO. Mask R-CNN (He et al., 2017) with the C4 backbone variant (Wu et al., 2019) was fine-tuned using the 1 schedule. AP metrics defined by COCO are reported here. For the linear classification task, Top-1 accuracy (in %) for Places205 (Zhou et al., 2014) and mAP for VOC07 (Everingham et al., 2010) are based on the frozen representations pre-trained on ImageNet. The best results are in **bold**.

| Methods | Object Detection | | | Instance Segmentation | | | Linear Classification | |
|---|---|---|---|---|---|---|---|---|
| | $AP^{bb}$ | $AP^{bb}_{50}$ | $AP^{bb}_{75}$ | $AP^{mk}$ | $AP^{mk}_{50}$ | $AP^{mk}_{75}$ | VOC2007 | Places205 |
| Sup. | 38.2 | 58.2 | 41.2 | 33.3 | 54.7 | 35.2 | 87.5 | 53.2 |
| MoCo-v2 | **39.3** | 58.9 | 42.5 | **34.4** | 55.8 | 36.5 | 86.4 | 51.8 |
| SwAV | 38.4 | 58.6 | 41.3 | 33.8 | 55.2 | 35.9 | 86.4 | 52.8 |
| SimSiam | 39.2 | **59.3** | 42.1 | **34.4** | **56.0** | **36.7** | - | - |
| BT | 39.2 | 59.0 | 42.5 | 34.3 | **56.0** | 36.5 | 86.2 | 54.1 |
| MUSIC (Ours) | **39.3** | 59.1 | **42.6** | **34.4** | 55.8 | 36.6 | **86.5** | **54.8** |

son methods in the same settings. MUSIC performs on par with the current methods and slightly better than Barlow Twins on the object detection and segmentation tasks. On the other hand, the linear classification results on VOC2007 and Places205 datasets show that MUSIC achieved better results than the selected methods. Also, similar to the other SSRL methods, MUSIC can effectively improve the downstream tasks in the transfer learning settings. All implementation details for the reproduction of transfer learning results are in Appendix G.2.

Table 3: **KNN classification**. Top-1 accuracy with 20 and 200 nearest neighbors are reported. The best results are highlighted in **bold**.

| Method | 20-NN | 200-NN |
|---|---|---|
| NPID (Wu et al., 2018) | - | 46.5 |
| LA (Zhuang et al., 2019) | - | 49.4 |
| PCL (Li et al., 2021) | 54.5 | - |
| BYOL (Grill et al., 2020) | 66.7 | **64.9** |
| SwAV (Caron et al., 2020) | 65.7 | 62.7 |
| BT (Zbontar et al., 2021) | 64.8 | 62.9 |
| VICReg (Bardes et al., 2022) | 64.5 | 62.9 |
| MUSIC (Ours) | **67.0** | **64.9** |

Table 4: **Batch Size.** Top-1 accuracy (in %) results for linear classification on ImageNet were obtained based on ResNet50 with 100 pre-training epochs. The best results are highlighted in **bold**.

| Batch Size | 512 | 1024 | 2048 | 4096 |
|---|---|---|---|---|
| SimSiam | 68.1 | 68.0 | 67.9 | 64.0 |
| VICReg | 68.2 | 68.3 | 68.6 | 67.8 |
| MUSIC | **68.3** | **69.3** | **69.4** | **68.7** |

**KNN Classification on ImageNet.** Another common protocol for evaluating representation learning methods is by K-Nearest-Neighbors (KNN) classification on ImageNet. We followed the recent studies (Wu et al., 2018; Zhuang et al., 2019; Caron et al., 2020; Bardes et al., 2022) that built KNN classifiers with the learned representations on the training set of ImageNet and evaluated the KNN classification results on the validation set of ImageNet. The results with 20 and 200 nearest neighbors are reported in Table 3, showing that MUSIC achieved the best performance among the comparison methods. Since the KNN classifier determines the class of a sample by directly searching its nearest samples in the feature space, the representation features learned by MUSIC is more semantically similar to each other among the nearest neighbors than those learned by other methods. Thus, MUSIC has the potential superiority when applied to the downstream tasks based on the nearest neighbors search.

All the above results demonstrate the effectiveness and superiority of MUSIC as a new embedding strategy in with the information-theoretic optimization framework. In the following subsections, the characteristics and superiority of MUSIC will be further discussed.

## 4.2 EMPIRICAL ANALYSIS

In this subsection, we comprehensively evaluate the proposed MUSIC method in various settings and compare it with other SSRL methods if the corresponding results in the same or comparable settings were already reported. All the models were evaluated with linear classification on ImageNet.

**Effect of Batch Size.** SSRL methods usually require a large batch size or a memory bank especially for contrastive learning. Here we evaluated MUSIC with different batch sizes and the results are reported in Table 4. It shows that MUSIC achieved consistently better results than the latest method VICReg over different batch sizes. As discussed in Subsection 2.4, an intrinsic property of MUSIC is to discriminatively encode different instances, making it work well without a large number of contrastive samples.

Table 5: **Training epochs**. Top-1 accuracy (in %) of linear classification on ImageNet using ResNet-50. The best results are highlighted in **bold** while the second best results are underlined.

| Methods | SimCLR | MoCo v2 | BYOL | SwAV | SimSiam | MUSIC |
|---------|--------|---------|------|------|---------|-------|
| 100 epochs | 66.5 | 67.4 | 66.5 | 66.5 | 68.1 | **69.4** |
| 200 epochs | 68.3 | 69.9 | 70.6 | 69.1 | 70.0 | **71.8** |
| 400 epochs | 69.8 | 71.0 | **73.2** | 70.7 | 70.8 | 73.1 |
| 800 epochs | 70.4 | 72.2 | **74.3** | 71.8 | 71.3 | 73.4 |

**Effect of Epoch Number.** The SSRL methods in different studies do not always use the same training epochs due to different computational costs and environments. MUSIC was evaluated with different numbers of training epochs as reported in Table 5. MUSIC is consistently better than most of the existing methods on all different training epochs. When the numbers of training epochs were small (100 and 200), MUSIC can converge to the best results.

Table 6: **Projector Depth**. The best results are highlighted in **bold**.

Table 7: **Feature Dimension**. The best Top-1 accuracies are highlighted in **bold**.

| Depth | 2 | 3 | 4 |
|-------|---|---|---|
| Top-1 | **69.4** | 68.5 | 67.9 |
| Top-5 | **89.3** | 88.3 | 87.9 |
| Time/100ep | 8.5h | 9.6h | 10.8h |
| Memory/GPU | 10.4G | 11.5G | 12.5G |

| $D_{\text{VICReg}}$ | 1024 | 2048 | 4096 | 8192 | 16384 |
|---------|------|------|------|------|-------|
| $D_{\text{MUSIC}}$ | 960 | 2000 | 4080 | 8160 | 16320 |
| VICReg | 62.4 | 65.1 | 67.3 | 68.6 | 68.8 |
| MUSIC | **64.1** | **66.6** | **69.2** | **69.4** | **69.1** |
| Time/100ep | 7.6h | 7.7h | 8.0h | 8.5h | 10.9h |
| Memory/GPU | 7.6G | 8.0G | 8.5G | 10.4G | 15.9G |

**Effect of Projector Depth.** The existing studies (Chen & He, 2021; Zbontar et al., 2021; Bardes et al., 2022) show that using a three-layer MLP as the projector achieved the best results. However, MUSIC has a different behavior that a two-layer MLP achieved the best results as shown in Table 6. It may be due to the discriminability and diversity of MUSIC embeddings, allowing it to learn informational representations more effectively. At the same time, the computational cost can be reduced, especially for the fully-connected MLP with high-dimensional inputs and outputs. The running time per 100 epochs and the peak memory per GPU for different projector depth are reported in Table 6, where the computational environment is described in Appendix E. Moreover, the comparison results of different methods in Appendix F show that MUSIC cannot only reduce the running time and memory cost but also achieves better performance than Barlow Twins and VICReg.

**Effect of Feature Dimension.** In the previous Barlow Twins and VICReg studies, it was found that a very high-dimensional embedding vector is necessary for improving the representation learning performance. For MUSIC, the feature dimension plays an important role as well. The results of different feature dimensions for VICReg and MUSIC are reported in Table 7, where the the dimensions of MUSIC embeddings are similar to those of VICReg embeddings while keeping the dimension of each segment the same, $D_S = 80$. It can be seen that MUSIC achieved consistently better results than VICReg on different embedding feature dimensions. Importantly, when the embedding feature dimension was reasonably large (4,096 and 8,192), MUSIC achieves the best results that are even better than the best results of VICReg using the larger dimension of 16,384. This is because that minimizing linear correlation by the existing methods cannot ensure the minimized

Table 8: **Loss terms**. The best results are highlighted in **bold**.

| Loss | DE+OE | OE+TI | DE+OE+TIC | DE+OE+TI |
|------|-------|-------|-----------|----------|
| Top-1 | 65.4 | 64.1 | 68.3 | **69.4** |
| Top-5 | 86.9 | 86.4 | 88.6 | **89.3** |

Table 9: **Segment Dimension**. The best results are highlighted in **bold**.

| $D_S$ | 32 | 64 | 80 | 96 | 128 |
|-------|-----|-----|-----|-----|-----|
| Top-1 | 67.8 | 69.1 | **69.4** | 69.2 | 68.4 |
| Top-5 | 88.5 | 89.1 | **89.3** | 89.1 | 88.5 |

non-linear dependency while MUSIC can minimize any form of dependency between any two feature variables. Therefore, the redundancy between MUSIC feature variables tends to be lower than the existing methods so that feature dimension can be reduced for even better results. In principle, the large embedding feature dimension (i.e., 16,384) significantly increases the computational and memory cost for Barlow Twins, VICReg, and MUSIC that compute the covariance or joint probability matrix, which was also discussed in the Barlow Twins study (Zbontar et al., 2021). This point is demonstrated in Table 7 by evaluating running time and memory cost, where the computational environment is described in Appendix E. Thus, MUSIC is both efficient and effective.

**Effect of Loss Function.** The effect of different loss terms was evaluated in Table 8, where DE, OE, TIC, and TI denote the diagonal entropy loss, off-diagonal entropy loss, transformation invariance loss implemented with cross-entropy, and transformation invariance loss implemented with inner-product, respectively. As described in Subsection 2.4, only optimizing the entropy loss (DE+OE) allows MUSIC to avoid trivial solutions and learn informational representations. This theoretical analysis is consistent with the empirical results in Table 8 that 65.4% Top-1 was achieved using the entropy loss only, comparable to some methods reported in Table 1. Adding the enhanced transformation invariance constraint in instance-level significantly improved the performance, as also discussed in the Subsection 2.3. Without adding the DE loss, the results were significantly degraded, as the DE loss not only enhances the transformation invariance but also helps learn non-trivial and complementary attributes in each and every segment. In our experiments, minimizing the cross-entropy degraded the performance compared with the inner-product implementation.

**Effect of Segment Dimension.** Finally, the effect of our unique hyperparameter, *i.e.*, segment dimension, was evaluated. Our empirical results with different segment dimensions in Table 9 indicate that $D_S = 80$ achieved the best results, where the dimension of the whole embedding vector was kept the same. It shows that the representation performance is not sensitive to this hyperparameter.

## 5 DISCUSSIONS AND CONCLUSION

**From pairwise independence to mutual independence.** Although MUSIC is minimizing the pairwise independence, the minimum mutual independence among multiple feature variables cannot be ensured (Gallager, 2013). In other words, redundancy may still exist among multiple feature variables. Similar to the study (Niu & Wang, 2022) that imposes high-order moment constraints on the embedding features, maximizing joint entropy among multiple variables could be implemented to reduce the redundancy further for even better self-learning performance.

**From representation learning to hierarchical clustering.** Each segment in MUSIC can be regarded as a clustering head, while different segments promote different and independent clustering criteria. In this study, we mainly focus on the general representation learning task, where each segment has the same number of clusters and every two segments are independent. This idea could be extended to hierarchical clustering; *e.g.*, different segments may have different numbers of hierarchical clusters and the independence constraint between segments can be adapted with task prior.

More discussions and comparisons between MUSIC and related methods are in Appendix D.

In conclusion, we have presented the multi-segment informational coding (MUSIC) scheme that discretizes feature variables and a new information-theoretic SSRL framework. Theoretical analysis ensures that the optimized embedding features are transform-invariant, non-trivial, diverse, and discriminative. Importantly, MUSIC can minimize any form of dependency between feature variables beyond the linear correlation in current methods. Various evaluation results have clearly shown the effectiveness and superiority of MUSIC in terms of both accuracy and efficiency. MUSIC could be adapted to more downstream tasks, such as clustering and dense prediction tasks.

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

## APPENDIX A    PYTORCH PSEUDOCODE

An exemplary implementation for MUSIC in the PyTorch-style is described in Algorithm 1.

---

**Algorithm 1:** PyTorch-style pseudocode for MUSIC

```
# f:  network function
# lambda:  weight on the transformation invariance loss term
# N: batch size
# D: dimensionality of the embedding vector
# D_S: dimensionality of each segment
# S=D/D_S: number of segments
#
# select:  select the diagonal elements of diagonal blocks and
 all elements of off-diagonal blocks

for x in loader:  # load a batch with N samples
  # two randomly augmented versions of x
  x', x'' = augment(x)

  # compute embeddings
  z' = f(x')
  z'' = f(x'')

  # multi-segmental informational coding
  x' = torch.reshape(x', [N, -1, D_S]) # N×S×D_S
  x'' = torch.reshape(x'', [N, -1, D_S]) # N×S×D_S
  q' = torch.softmax(x', dim=2) # softmax normalization
  q'' = torch.softmax(x'', dim=2) # softmax normalization

  # compute transformation invariance loss
  loss_TI = -torch.log((q'*q'').sum(dim=2)).mean()

  # compute entropy loss
  q' = torch.reshape(q', [N, D]) # N × D
  q'' = torch.reshape(q'', [N, D]) # N × D
  P = torch.einsum('np,nq->pq', [q', q'']) / N # compute
   empirical joint probability distribution
  P_s = select(P)
  loss_ent = (P_s * torch.log(P_s)).sum() / (S × S)

  # final loss
  loss = loss_ent + lambda * loss_TI # lambda=1 by default

  # optimization step
  loss.backward()
  optimizer.step()
```

---

## APPENDIX B    THEORETICAL ANALYSIS

**The optimal solution to the first part in $L_{ent}$.** As described in Subsection 2.4, the entropy loss function consists of two parts: (1) the entropy over diagonal elements of diagonal blocks and (2) the entropy over all elements of off-diagonal blocks, as illustrated in Fig. 2. Now, let us minimize the first part, which is formulated as

$$L_{ent} = \frac{1}{S} \sum_{s',s'',s'=s''} \sum_{d',d'',d'=d''} \boldsymbol{P}(s',s'',d',d'') \log(\boldsymbol{P}(s',s'',d',d'')) \qquad \text{(B-7)}$$

Since every diagonal block has the same optimal solution, we only need to consider the $s^{th}$ diagonal block, and the objective function can be simplified as

$$L_{ent}(s,s) = \sum_{d=1}^{D_S} \boldsymbol{P}(s,s,d,d) \log(\boldsymbol{P}(s,s,d,d)) \tag{B-8}$$

where $0 \leq \boldsymbol{P}(s,s,d,d) \leq 1, 0 \leq \sum_{d=1}^{D_S} \boldsymbol{P}(s,s,d,d) \leq 1$. Then, it is easy to find the solution that minimizes this objective function; i.e., $\forall s,d, \boldsymbol{P}(s,s,d,d) = \frac{1}{D_S}$. Thus, $\forall s,d$, we have

$$\sum_d \boldsymbol{P}(s,s,d,d) = \sum_{d=1}^{D_S} \frac{1}{D_S} = 1 \tag{B-9}$$

As defined in Eqs. (1) and (2), we have $\forall s,d, 0 \leq \boldsymbol{q}_i'(s,d) \leq 1, 0 \leq \boldsymbol{q}_i''(s,d) \leq 1, \sum_{d=1}^{D_S} \boldsymbol{q}_i'(s,d) = 1, \sum_{d=1}^{D_S} \boldsymbol{q}_i''(s,d) = 1$, and $\boldsymbol{P}(s,s,d,d) = \frac{1}{N} \sum_{i=1}^{N} \boldsymbol{q}_i'(s,d) \boldsymbol{q}_i''(s,d)$.

Given the above conditions, let us next prove that for $\forall s, \exists d, \boldsymbol{q}_i'(s,d) = \boldsymbol{q}_i''(s,d) = 1$ by contradiction.

If its negation is true, i.e., $\forall s,d$, either $0 \leq \boldsymbol{q}_i'(s,d) < 1$ or $0 \leq \boldsymbol{q}_i''(s,d) < 1$, then we have $\forall s,d$, either $\boldsymbol{q}_i'(s,d) < \sum_{d'=1}^{D_S} \boldsymbol{q}_i'(s,d') = 1$, or $\boldsymbol{q}_i''(s,d) < \sum_{d''=1}^{D_S} \boldsymbol{q}_i''(s,d'') = 1$. For $\boldsymbol{q}_i''(s,d) < \sum_{d''=1}^{D_S} \boldsymbol{q}_i''(s,d'') = 1$, we have

$$\begin{aligned}
\sum_{d=1}^{D_S} \boldsymbol{P}(s,s,d,d) &= \sum_{d=1}^{D_S} \frac{1}{N} \sum_{i=1}^{N} \boldsymbol{q}_i'(s,d) \boldsymbol{q}_i''(s,d) \\
&= \frac{1}{N} \sum_{i=1}^{N} \sum_{d=1}^{D_S} \boldsymbol{q}_i'(s,d) \boldsymbol{q}_i''(s,d) \\
&< \frac{1}{N} \sum_{i=1}^{N} \sum_{d=1}^{D_S} \left( \boldsymbol{q}_i'(s,d) \sum_{d''}^{D_S} \boldsymbol{q}_i''(s,d'') \right) \\
&= 1
\end{aligned} \tag{B-10}$$

That is, $\sum_d \boldsymbol{P}(s,s,d,d) < 1$, which leads to a contradiction with Eq. (B-9). Similarly, we have the same contradiction for $\boldsymbol{q}_i'(s',d) < \sum_{d'=1}^{D_S} \boldsymbol{q}_i'(s',d') = 1$. Therefore, the statement that $\forall s, \exists d, \boldsymbol{q}_i'(s,d) = \boldsymbol{q}_i''(s,d) = 1$ is true. It means that for $\forall s, \boldsymbol{q}_i'(s,:)$ and $\boldsymbol{q}_i''(s,:)$ are one-hot vectors and equal to each other.

Because $\forall s,d, \boldsymbol{P}(s,s,d,d) = \frac{1}{D_S}, \boldsymbol{q}'(s,d) = \boldsymbol{q}''(s,d)$, and $\boldsymbol{q}'(s,:)$ and $\boldsymbol{q}''(s,:)$ are one-hot vectors, then $\boldsymbol{P}(s,s,d,d) = \frac{1}{N} \sum_{i=1}^{N} \boldsymbol{q}_i'(s,d) \boldsymbol{q}_i''(s,d) = \frac{1}{N} \sum_{i=1}^{N} \boldsymbol{q}_i'(s,d) = \boldsymbol{p}(s,d) = \frac{1}{D_S}$.

**Covariance of the optimal solution.** The optimal solution to maximize the joint entropy over the off-diagonal blocks for the second part is $\forall s' \neq s'', \boldsymbol{P}(s',s'',d',d'') = \mathbb{E}[\boldsymbol{q}(s',d')\boldsymbol{q}(s'',d'')] = \frac{1}{(D_S)^2}$. According to the above proof, we have $\forall s,d, \boldsymbol{p}(s,d) = \mathbb{E}[\boldsymbol{q}(s,d)] = \frac{1}{D_S}$. We can theoretically demonstrate that the covariance is zero between any two units from different segments. Specifically, $\forall s',s'',d',d'',s' \neq s''$, we have

$$\begin{aligned}
\mathrm{cov}[\boldsymbol{q}(s',d'),\boldsymbol{q}(s'',d'')] &= \mathbb{E}[\boldsymbol{q}(s',d')\boldsymbol{q}(s'',d'')] - \mathbb{E}[\boldsymbol{q}(s',d')]\mathbb{E}[\boldsymbol{q}(s'',d'')] \\
&= \frac{1}{(D_S)^2} - \frac{1}{D_S} \times \frac{1}{D_S} = 0.
\end{aligned} \tag{B-11}$$

Since $\forall s,d',d'', \sum_{d',d''} \boldsymbol{P}(s,s,d',d'') = 1$ and $\forall s,d',d'',d' = d'', \boldsymbol{P}(s,s,d',d'') = \frac{1}{D_S}$, then $\forall s,d',d'',d' \neq d'', \boldsymbol{P}(s,s,d',d'') = 0$. We can demonstrate that any two units within the same segment are negatively correlated. Formally, $\forall s,d',d'',d' \neq d''$, we have

$$\begin{aligned}
\mathrm{cov}[\boldsymbol{q}(s,d'),\boldsymbol{q}(s,d'')] &= \mathbb{E}[\boldsymbol{q}(s,d')\boldsymbol{q}(s,d'')] - \mathbb{E}[\boldsymbol{q}(s,d')]\mathbb{E}[\boldsymbol{q}(s,d'')] \\
&= \boldsymbol{P}(s,s,d',d'') - \boldsymbol{p}(s,d')\boldsymbol{p}(s,d'') \\
&= 0 - \frac{1}{D_S} \times \frac{1}{D_S} = -\frac{1}{D_S^2}.
\end{aligned} \tag{B-12}$$

That is, every unit within each segment encodes discriminative and complementary features, while the units from different segments encode unrelated and diverse features.

APPENDIX C    VISUALIZATION OF MUSIC EMBEDDING

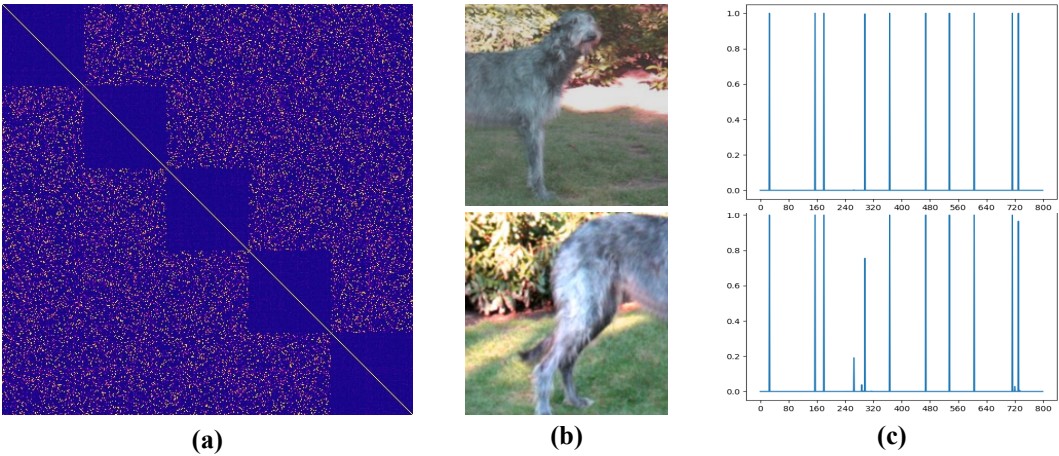

**(a)**       **(b)**       **(c)**

Figure C.3: Visualization of MUSIC elements. (a) A partial joint probability matrix, where blue and yellow respectively represent small and large values, (b) two transformations of the same image, and (c) partial embedding vectors corresponding to the images in (b).

In Fig. C.3, we visualize the MUSIC elements including an empirical joint probability matrix and individual embeddings, where the empirical joint probabilities were computed over the whole ImageNet train dataset and only the left-upper partial matrix of $400 \times 400$ and the first 800 units of embedding features are selected for visualization. The theoretical analysis in Subsection 2.4 demonstrates that the embedding statistics are enforced to be uniform; *i.e.*, $\forall s, d, \boldsymbol{P}(s, s, d, d) = \frac{1}{D_S}$, meaning that the probabilities of the diagonal elements in all diagonal blocks are equal and those of off-diagonal elements are zeros. Also, the probabilities of all elements of the off-diagonal blocks are equal, *i.e.*, $\forall s', s'', d', d'', s' \neq s'', \boldsymbol{P}(s', s'', d', d'') = \frac{1}{(D_S)^2}$. The empirical joint probability matrix visualized in Fig. C.3-(a) is consistent with theoretical analysis although not a perfect match. Furthermore, Figs. C.3-(b) and (c) show that the embedding features in each segment tend to be one-hot and invariant to the transformations, which are also consistent with the theoretical analysis.

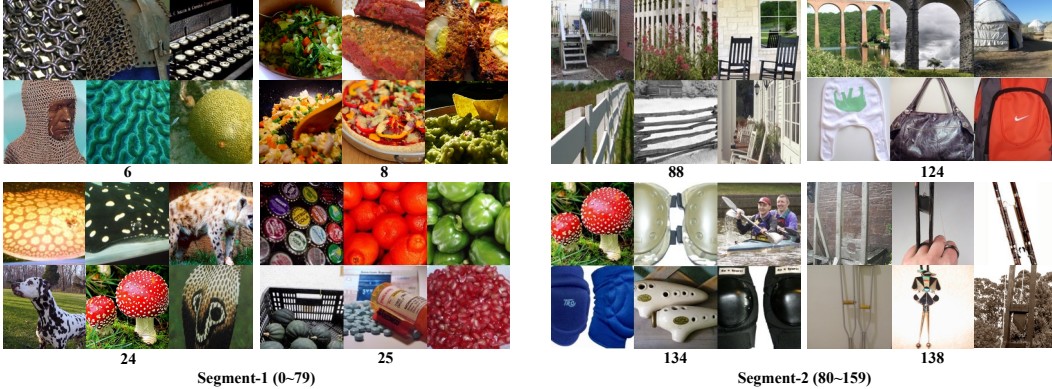

Figure C.4: Visualization of learned MUSIC features on ImageNet validation set. The left side shows the samples assigned to the features indexed by 6, 8, 24, and 25 in the first segment. The right side shows the samples assigned to the features indexed by 88, 124, 134, and 138 in the second segment.

To qualitatively evaluate if meaningful embedding features are learned, in Fig. C.4 we show some examples assigned to specific units in the first two segments, where the whole embedding vector has 8,160 units including 102 segments, and each segment has 80 units. Specifically, some features in

the first segment represent different types of textures; *e.g.*, the feature unit indexed by 24 represents the dot style textures, and the units 6, 8, and 25 correspond to other specific textures/patterns. In the second segment, some features represent different shapes; *e.g.*, the unit 124 abstract a "∩" shape, and the units 88, 134, 138 represent other shapes/patterns. Obviously, the first and second segments use different principles to group samples; *e.g.*, the image containing red mushrooms in the first segment is grouped (indexed by 24) with the objects having similar textures, while in the second segment it is grouped (indexed by 134) with the images having twin/repeated objects. These visual results indicate that the learned MUSIC embedding features are indeed meaningful and consistent to the general properties of Fig. 1, which are ensured by the theoretical analysis.

## APPENDIX D    RELATED WORK

For self-supervised representation learning (SSRL), various pretext tasks were designed such as denoising auto-encoders (Vincent et al., 2008), context auto-encoders (Pathak et al., 2016), colorization and cross-channel auto-encoders (Zhang et al., 2016; 2017), masked auto-encoders (He et al., 2022), rotation (Gidaris et al., 2018; 2020), patch ordering (Noroozi & Favaro, 2016; Doersch et al., 2015; Chen et al., 2021), clustering (Caron et al., 2018; 2019; Asano et al., 2019; Yan et al., 2020; Huang et al., 2019; Zhuang et al., 2019; Gidaris et al., 2021), and instance discrimination (Dosovitskiy et al., 2014; Wu et al., 2018; Tian et al., 2020b; Ye et al., 2019; Dwibedi et al., 2021). Here we compare MUSIC with the most related SSRL methods, including contrastive learning, asymmetric non-contrastive Learning, clustering-based SSRL, dense SSRL, and the more recent non-asymmetric and non-contrastive learning methods.

**Contrastive Learning**.  The contrastive learning based SSRL methods (Chen et al., 2020a; He et al., 2020) need to directly compare the features between negative pairs. Thus, large batch sizes are required, such as for SimCLR (Chen et al., 2020a). MoCo (He et al., 2020) uses a memory bank to store a large number of features as negative samples so that small batch sizes can be used, while it requires a momentum updating technique. The theoretical analysis for contrastive learning is based on estimating the lower bound on mutual information between different views (Oord et al., 2018). In contrast, MUSIC doesn't need negative samples, memory bank, or momentum updating, while it can still discriminate instances. Without directly comparing a large number of negative pairs for instance discrimination, MUSIC naturally encodes different instances with different embeddings via maximizing the joint entropy, as demonstrated in our theoretical analysis. MUSIC can directly use information measurements for both optimization and analysis instead of estimating the lower bound.

**Asymmetric non-contrastive Learning**.  BYOL (Richemond et al., 2020) and SimSiam (Chen & He, 2021) demonstrate that meaningful representations can be learned without using negative pairs. However, these methods depend on asymmetric architectures and stop gradient techniques to avoid trivial solutions.  The follow-up theoretical analysis (Wang & Isola, 2020; Zhang et al., 2021a; Richemond et al., 2020; Tian et al., 2021) leverage various concepts under different assumptions to demonstrate why these methods can avoid trivial solutions. MUSIC requires neither negative samples nor asymmetric designs. Moreover, MUSIC enables a new information-theoretic SSRL framework where the information measures are directly used for both numerical optimization and theoretical analysis, which intrinsically avoids trivial solutions and learns meaningful features.

**Clustering-based SSRL**. DeepCluster (Caron et al., 2018) iteratively performs clustering with an extra kmeans algorithm on the features extracted in the previous step, and updates the weights of the network using the cluster assignments as supervision. To avoid trivial solutions, random samples are selected for the empty cluster to compute the centroid. It is time consuming for kmeans to cluster the whole large datasets, and the random selected samples is hard to form a meaningful cluster. Similarly, SELA (Asano et al., 2019) leverages the Sinkhorn-Knopp algorithm to iteratively perform clustering and optimize the clustering networks with the assigned cluster labels in an online manner. SwAV (Caron et al., 2020) alternatively computes the cluster assignment of one view and optimize the network to predict the same assignment for other views of the same sample. As a contrastive learning method, SwAV still requires a lot of prototype vectors for negative comparisons between embeddings and codes. In can be seen that all these clustering methods require to compute a large number of extra cluster centers and leverage extra algorithms to compute assignments. From the clustering perspective, each segment in the MUSIC embedding can be regarded as

a cluster assignment, and multiple segments can be regarded as performing multiple clustering simultaneously, where different segments have different clustering principles. Different from current clustering based methods, MUSIC does not require computing a large number of cluster centers or an extra algorithm to estimate the cluster assignments iteratively.

**Dense SSRL**. Some excellent studies, such as DenseCL (Wang et al., 2021) and DenseSiam (Zhang et al., 2022), design self-supervised/unsupervised learning methods for improving dense prediction tasks, including object detection (Tian et al., 2019), instance segmentation (Zang et al., 2021), and semantic segmentation (Zhang et al., 2021b). The basic idea is to enhance the pixel-level and region-level consistency in the self-supervised/unsupervised learning setting. DenseCL (Wang et al., 2021) proposes to optimize a pairwise contrastive (dis)similarity loss at the pixel level between two views of input images. Most recently, DenseSiam (Zhang et al., 2022) proposes to optimizes the consistency of different levels based on the simple Siamese network without needing negative pixel pairs, momentum encoders or heuristic masks. Synergistically, MUSIC can be also used for dense prediction tasks by discretizing the feature vector of each sub-region/pixel and learning dense representations in the information-theoretic framework.

**Non-asymmetric and non-contrastive learning**. Recently, WMSE (Ermolov et al., 2021), Barlow Twins Zbontar et al. (2021), and VICReg (Bardes et al., 2022) propose to train a simple twin network architecture using covariance matrix based loss functions without needing any asymmetric or constrastive learning techniques. Specifically, WMSE (Ermolov et al., 2021) proposes to minimize the MSE distance between different views and enforce the self-covariance matrix to be an identity-matrix. Barlow Twins (Zbontar et al., 2021) optimizes the cross-covariance matrix to be an identity-matrix. VICReg (Bardes et al., 2022) proposes three loss terms including invariance, variance, and covariance. The theoretical analysis for Barlow Twins (Zbontar et al., 2021) assumes the Gaussian distribution assumption of embedding features, then the loss function can be interpreted with the information measures under some approximations. One of the key ideas of these methods is to reduce the redundancy between feature variables by minimizing the linear correlation. In contrast, MUSIC discretizes the feature variables making the probability distribution estimable so that the information-measures (entropy/mutual information/(in)dependence) can be directly used for both numerical optimization and theoretical analysis without assuming the Gaussian distribution. Importantly, our theoretical analysis shows that MUSIC is exactly minimizing the mutual information or any dependence between feature variables, which is beyond the linear correlation constraints used in current methods. That is why MUSIC can use a shorter embedding vector and achieve even better results at a lower computational cost.

## APPENDIX E  COMPUTATIONAL ENVIRONMENT

MUSIC models were distributively trained on four nodes, each of which has the system information:

- $2\times$ 20 core 2.5 GHz Intel Xeon Gold 6248;
- $8\times$ NVIDIA Tesla V100 GPU each with 32 GiB HBM;
- 768 GiB RAM per node;
- Dual 100 Gb EDR Infiniband

## APPENDIX F  RUNNING TIME

In Table 10, the computational cost of MUSIC was evaluated and compared with other methods. All methods were run on 32 Tesla V100 GPUs. These methods offer different trade-offs among running time, memory and performance. SwAV with multi-crop and BYOL achieve better performance at the additional computational cost and memory usage. Barlow Twins and VICReg have balanced results with less memory than BYOL and SwAV (multi-crop), faster speed than SwAV (multi-crop), but a slightly worse performance. Compared with the most related Barlow Twins and VICReg methods, MUSIC cannot only reduce the running time and memory usage significantly, but also improve the performance. It is due to that MUSIC can use a shallower fully-connected MLP head for a better performance as discussed in Subsection 4.2. The computational cost of MUSIC will be significantly reduced further when using a ($\times2$) lower dimension for embeddings, and the performance would be degraded very slightly, as discussed in Subsection 4.2.

Table 10: **Running time and peak memory**. Comparison of different methods in terms of the running time over 100 epochs, the peak memory on a single GPU, and the top-1 accuracy (%) on linear classification on top of the frozen representations. All models were distributively trained on 32 Tesla V100 GPUs.

| Method | Time / 100epochs | Peak memory / GPU | Top-1 accuracy |
|---|---|---|---|
| SwAV | 9h | 9.5G | 71.8 |
| SwAV (w/multi-crop) | 13h | 12.9G | 75.3 |
| BYOL | 10h | 14.6G | 74.3 |
| Barlow Twins | 12h | 11.3G | 73.2 |
| VICReg | 11h | 11.3G | 73.2 |
| MUSIC | 8.5h | 10.4G | 73.6 |

## APPENDIX G    IMPLEMENTATION DETAILS

### G.1    EVALUATION ON IMAGENET

**Linear classification:** For all evaluation experiments on ImageNet linear classification, we followed the standard procedure that a linear classifier was trained on top of the frozen backbone of a ResNet-50 pre-trained with MUSIC. The SGD optimizer was used with a learning rate of 0.02, a cosine decay, a weight decay of $10^{-6}$, a batch size of 256, and 100 training epochs. In the training stage, the images were augmented by the composition of random cropping and resizing of ratio 0.2 to 1.0 for size 224×224, and random horizontal flips. In the testing stage, the images were simply cropped from the image center and resized to $224 \times 224$.

**Semi-supervised learning:** In the semi-supervised learning setting, a linear classifier was appended to the pre-trained backbone with MUSIC, and the network was fine-tuned using 1% and 10% of the labels. The SGD optimizer was used with no weight decay and a batch size of 256, and the model was trained for 20 epochs. In the 1% of labels case, we used a learning rate of 0.08 for the encoder and 0.1 for the linear head. In the 10% of labels case, we used 0.02 for the encoder and 0.1 for the linear head. Both these learning rates followed a cosine decay schedule. The data augmentation steps for training and testing followed the same settings of the linear evaluation.

### G.2    TRANSFER LEARNING

**Object detection and instance segmentation:** Mask R-CNN  (He et al., 2017) with the C-4 backbone was trained on the COCO 2017 train split and tested on the validation set. We used a learning rate of 0.1 and kept the other parameters the same as in the 1 schedule in detectron2.

**Linear classification:** We followed the exact settings from PIRL  (Misra & Maaten, 2020) in evaluating linear classifiers on the Places-205 and VOC07 datasets. For Places-205, a linear classifier was trained using the SGD optimizer for 14 epochs with a learning rate of 0.01 reduced by a factor of 10 at epochs 5 and 10, a weight decay of $5 \times 10^{-4}$, and a momentum of 0.9. For VOC2007 dataset, we trained SVM classifiers, where the $C$ values were computed using cross-validation.

Table 11: Top-1 linear classification accuracies on CIFAR10. Here the embedding dimension for all models was set to 1024. The MUSIC results with different segment dimensions are reported.

| Models | Barlow Twins | VICReg | MUSIC | | | | | |
|---|---|---|---|---|---|---|---|---|
| | | | 8 | 16 | 24 | 32 | 48 | 64 |
| TOP-1 | 88.2 | 88.5 | 88.5 | 88.6 | 88.6 | **89.2** | **89.2** | 88.6 |

## APPENDIX H    RESULTS ON CIFAR10

Here we further evaluated the characteristics of MUSIC in terms of the segment dimension on the CIFAR10 dataset. Without using any asymmetric or contrastive techniques, Barlow Twins and VICReg are regarded as the baseline methods for MUSIC. Specifically, the batch size was 512 and

the total dimension of the embedding vector was 1,024, the base learning rate was 0.1, the number of training epochs was 800, and all other hyper-parameters were kept the default settings for MUSIC, Barlow Twins, and VICReg. Top-1 accuracy results are reported in Table 11, showing that the segment dimension should be adjusted according to the target dataset, which is similar to that the network architecture and the total feature dimension are usually associated with the scale and complexity of target datasets. Nevertheless, MUSIC achieved superior results than the baseline models over a large range of segment dimensions under the same evaluation setting.

