# OpenReview forum: "Multi-Segmental Informational Coding for Self-Supervised Representation Learning"
_ICLR.cc/2023/Conference — Submitted to ICLR 2023_

### Official Review · Reviewer_f3VU · 2022-10-22

**Confidence:** 2
**Correctness:** 2
**Technical Novelty And Significance:** 3
**Empirical Novelty And Significance:** 3
**Recommendation:** 6

**Clarity, Quality, Novelty And Reproducibility:**

The paper is very clear and understandable. It is well written in good English and the reported results are encouraging but not optimal. The authors have provided pseudocodes in the supplementary material which I expect to be useful for the reproducibility of the method. Therefore, I think the proposed method has a lot of scope, but proper evidence and justifications are missing in some cases.

**Strength And Weaknesses:**

**Strengths**

(1) The proposed method is based on an interesting idea of segmenting representation and estimating the probability distribution over discrete attributes, which I think is relatively new.

(2) The paper has considered standard benchmarks for experimentations to demonstrate the effectiveness of the proposed method, where the MUSIC model has performed consistently.

(3) The paper is theoretically well grounded and the authors have shown that the optimal MUSIC embedding features have zero covariance between any two features in different segments and negative covariance between the features within the same segment.

**Weaknesses**

(1) The proposed method have achieved minor improvement on very few datasets, whereas on the other datasets, the method couldn't surpass the state-of-the-art methods, which fails to justify the superiority of the proposal.

(2) The method is nicely motivated by the idea of multi-segmental discretisation, but the evidence that the learnt representation follows intuition is largely missing. At the moment, there is only one example in appendix C, but more such visualisation would have been useful and interesting.

(3) It is not clear or evident what information those individual segments are learning. Also the number of segments should be correlated with datasets and also on the size of crops, which have not been ablated in the paper at this stage and decided in an adhoc manner.

**Summary Of The Paper:**

This paper proposes multi-segmental informational coding as a new embedding scheme for self-supervised representation learning which divides an embedding vector into multiple segments to represent different types of attribute, and as a result each segment automatically learns a set of discrete and complementary attributes. The proposed model enables the estimation of the probability distribution over discrete attributes and thus the learning process can be directly guided by information measurements, reducing the feature redundancy beyond the linear correlation.

**Summary Of The Review:**

I have found the paper very interesting. The paper is theoretically justified. Some experimental or evidential issues evidences are there (mentioned in the weaknesses section) should be improved.

---

> ### Author Response · Authors · 2022-11-15
> **Response to Reviewer f3VU**
>
> We would like to thank reviewer $\textcolor{green}{f3VU}$ for the valuable feedback. Per your comments, we have provided more details on and further justification for MUSIC.
>
> **Q1**: _The proposed method has achieved minor improvement on very few datasets, whereas on the other datasets, the method couldn't surpass the state-of-the-art methods, which fails to justify the superiority of the proposal._
>
> **A1**: Although MUSIC does not surpass all current methods in all settings, it does achieve superior results compared with SOTA methods in various cases as summarized below and in the revision. Note that Barlow Twins and VICReg can be regarded as the baseline models to evaluate MUSIC, all of which do not use any asymmetric and contrastive techniques.
>
> (1) In Table 1, MUSIC achieves better linear classification than Barlow Twins and VICReg.
>
> (2) In Table 2, MUSIC achieves the best results in most cases for transfer learning tasks (object detection, instance segmentation, and linear classification), where the evaluation exactly follows Barlow Twins.
>
> (3) In Table 3, MUSIC achieves the best KNN classification on ImageNet among all methods including BYOL, SwAV, Barlow Twins, VICReg, etc., improving accuracy by 2+% over Barlow Twins and VICReg, which is a significant improvement.
>
> (4) In Table 5, MUSIC achieves superior results using different epochs, where the competing methods are SimCLR, MoCo v2, BYOL, SwAV, and SimSiam. These methods were evaluated in the same settings as described in the related papers.
>
> (5) In Table 7, MUSIC achieves consistently better results than VICReg using different feature dimensions.
>
> (6) In Table 10, MUSIC achieves better results in terms of both accuracy and efficiency than Barlow Twins and VICReg.
>
> **Q2**: _The method is nicely motivated by the idea of multi-segmental discretisation, but the evidence that the learnt representation follows intuition is largely missing. At the moment, there is only one example in appendix C, but more such visualisation would have been useful and interesting._
>
> **A2**: Two general properties are desired behind the intuition in Fig. 1: 1) within each segment, samples can be classified into a set of different and discrete attributes; and 2) different segments represent different classification principles, which means that the mutual information between different segments is minimized, or equivalently the information/entropy is maximized. Both these two general properties are justified in our theoretical analysis. Specifically, we have demonstrated that the optimized embeddings with the entropy loss function are associated with the following characteristics: 1) each segment is one-hot encoded and samples are evenly assigned into all discrete units within each segment; and 2) the mutual information between different segments approaches zero.
>
> Per the reviewer’s advice, we have visualized representative samples in terms of the learned MUSIC embeddings in Appendix C, showing that meaningful features, such as different textures, shapes, and patterns, are indeed learned in the different segments as intended. We also provided the source code for visualization in the supplementary material.
>
> **Q3**: _It is not clear or evident what information those individual segments are learning. Also, the number of segments should be correlated with datasets and also on the size of crops, which have not been ablated in the paper at this stage and decided in an ad hoc manner._
>
> **A3**: As discussed above, the theoretical analysis ensures that different segments learn different principles to discriminate samples, and representative examples have been augmented as visualized in Appendix C in the revision.
>
> We agree with the reviewer that the number of segments or the dimension of each segment (one can be determined by the other, given the fixed feature dimension) is correlated with datasets, similar to the situations where the total feature dimension and network architecture usually vary with datasets for representation learning. In Tables 7 and 9, we have studied the effects of these two factors on the ImageNet dataset. This is the single extra hyperparameter introduced by MUSIC. To address the reviewer’s concern, we have further applied MUSIC to the CIFAR10 dataset. The results with different segmentation dimensions are given below, which is consistent with the discussion above. MUSIC also achieves better results than Barlow Twins and VICReg over a large range of segment dimensions in the same evaluation setting. These new results have been added and discussed in Appendix H. The source codes for these results are provided in the supplementary material.
>
> |Models | Barlow Twins | VICReg ||||MUSIC|||
> |:---:|:---:|:---:|:---:|:---:|---:|:---:|:---:|:---:|
> | | | |8|16|24|32|48|64|
> |Top-1|88.2|88.5|88.5|88.6|88.6|**89.2**|**89.2**|88.6|

---

> > ### Author Response · Authors · 2022-12-12
> > **Follow up before the discussion is closed**
> >
> > Thank you again for your feedback. As we are approaching the end of the discussion period, we would like to ask whether our revision and response have addressed your initial concerns about our work. We are more than happy to clarify and discuss more should you have additional questions.

---

### Official Review · Reviewer_ZcT3 · 2022-10-24

**Confidence:** 4
**Correctness:** 3
**Technical Novelty And Significance:** 2
**Empirical Novelty And Significance:** 2
**Recommendation:** 5

**Clarity, Quality, Novelty And Reproducibility:**

The clarity and quality of the work are generally good, but with a few unclear statements that need further clarification, as mentioned above in the Weaknesses part. The novelty is a bit limited, considering similar prior works and the high-level idea of optimizing mutual information (or entropy). This work should be easy to be reproduced, as the PyTorch-style pseudo-code has been provided.

**Strength And Weaknesses:**

## Strengths

\+ The proposed idea of minimising the mutual information between different attributes is interesting.

\+ An analysis section (Sec. 2.4) was presented to help better understand the proposed method.

\+ An empirical analysis (Sec. 4.2) was provided to evaluate the components within the proposed method.

\+ The paper is in general well written and easy to follow.


## Weaknesses

\- The proposed method seems to be a disentangle approach for the projected feature vector, which was defined to be within several semantic categories (i.e. the segments as illustrated in Fig. 1). But it is unclear how it was guaranteed to have these semantic meanings for a projected vector. The evenly splitting of the vector was also not well justified.
On the other hand, although the projected vector was divided into several segments/attributes, the whole vector is in principle representing the same feature as without such partition. It is unclear, as a result, why this would introduce the claimed attributes information into the representation learning. The experimental results also did not suggest any noticeable improvement over previous works in representation learning.
It would be better if some examples/qualitative results could have been shown to validate that the claimed segments were actually related to the particular semantics.

\- The novelty and contributions of the proposed method are a bit limited. Considering that the general idea and framework are quite similar to prior works on self-supervised representation learning (e.g. SimCLR, BYOL, Barlow twins to name a few), the only difference is the final projected feature processing and the following loss to supervise the network training. Whereas apart from the segments part, even the entropy maximization is quite similar to the approach proposed in Barlow twins and VICReg in a high-level idea.

\- It is a bit unclear how the "theoretical analysis guarantees" the proposed model learns non-trivial, diverse and discriminative features, especially the discriminative features.

\- The performance of the proposed method is only comparable to several prior works and even worse than some of them (e.g. BYOL, Barlow Twins, VICReg). It is hard to tell and be convinced of the effectiveness of the proposed method and its significance.

\- The authors discussed the idea of hierarchical clustering in the last section, which is nice to see. This also relates the proposed method to the clustering-based approaches. It would be better to discuss and differentiate the proposed segment-based idea and those clustering-based ideas (e.g. DeepCluster, SwAV etc.) for visual representation learning, especially what is the benefit of the proposed method over those approaches.

**Summary Of The Paper:**

This paper presented a new method for self-supervised image representation learning. Specifically, a multi-segmental informational coding (MUSIC) scheme was proposed to achieve a new feature embedding, which divided the projected feature vector into multiple segments to represent different attributes. The loss function was then designed to maximize the entropy between segments and units within them. Experimental analysis on a few public datasets over benchmark vision tasks shows the effectiveness of the proposed method. The main contribution is the proposed MUSIC coding scheme.

**Summary Of The Review:**

The main idea of this paper is interesting and seems to make sense in self-supervised visual representation learning. But it was not well presented with sufficient evidence to justify the claims (the segments and attributes part). The experimental performance was also insufficient to support the claimed effectiveness of the proposed method. The overall technical novelty and contributions are also a bit limited.
As a result, this paper is considered to be not ready for publication. But the authors are encouraged to provide more details and justifications of the proposed method and the claimed contributions and re-submit to a future venue.

---

> ### Author Response · Authors · 2022-11-15
> **Response to ZcT3**
>
> We would like to thank reviewer $\textcolor{orange}{ZcT3}$ for the valuable feedback. Per your comments, we have provided more details and justifications, and clarified the contributions in the revision.
>
> **Q1: Regarding the MUSIC embedding**
>
> A1: MUSIC discretizes each feature variable with a segment of one-hot vector, and the whole embedding space consists of multiple segments of one-hot vectors. Before MUSIC, the feature variable is continuous in current embeddings. Since the distribution probability of continuous variables can be hardly estimated given finite samples, the information measures (entropy/mutual information/independence) defined on probability distributions cannot be directly computed. Discretization enables the estimation of the probability distribution over discrete units of each segment, thus information measures can be directly computed for both numerical optimization and theoretical analysis. Therefore, MUSIC presents a new embedding scheme and a new information-theoretic framework for self-supervised learning.
>
> **Q2: Regarding the non-trivial, diverse, and discriminative**
>
> A2:  **Non-trivial**: The optimized uniform distribution of samples over discrete attributes means that all samples cannot have the same embedding.
>
> **Diverse**: The minimized mutual information between segments means that different segments encode different types of features, or multiple segments cluster samples according to diverse criteria.
>
> **Discriminative**: The maximized joint entropy means that any two units from any two different segments have the equal possibility to co-occur; that is, samples are evenly assigned to all possible embeddings. Since the number of all possible embeddings is much larger ( $80^{102}$ vs 2,048) than the batch size, it will be enforced to encode different instances with different embeddings. Like contrastive learning, it also ensures non-trivial solutions.
>
> **Q3:  Regarding what features are learned**
>
> A3: There are two generally desirable properties in Fig. 1: 1) Within each segment samples can be clustered into a set of discrete groups; and 2) different segments represent different clustering criteria, meaning that the mutual information/redundancy between different segments is minimized, or equivalently the information/entropy is maximized. These two properties are demonstrated in our theoretical analysis.
>
> We have visualized representative results in Appendix D in the revision, showing that meaningful features, such as textures, shapes, and patterns, are indeed learned. We have also provided the source code for inspecting more features.
>
> **Q4: Regarding the novelty and contributions**
>
> A4: We have elaborated our contributions in the revision, especially the following two key points:
>
> * MUSIC is a new embedding scheme with discretized features and allows a new information-theoretic self-supervised learning framework. Information measures can be directly used for both numerical optimization and theoretical analysis, making the analysis more straightforward than the current methods that estimate the lower bound of mutual information or rely on other assumptions.
>
> * While the existing methods typically use linear correlation to minimize the redundancy between feature variables, MUSIC enforces independence beyond the linear correlation. That is why MUSIC can use a shorter vector and yet achieve better performance at a lower computational cost.
>
> **Q5: Comparison between MUSIC and related work**
>
> A5: A detailed comparison between MUSIC and related methods has been added in Appendix D.
>
> **Q6: Regarding the results**
>
> A6: MUSIC is either superior or comparable to current methods in common settings, as summarized below. The more recent methods, Barlow Twins and VICReg, can be regarded as the baseline models to evaluate MUSIC, all of which do not use any asymmetric or contrastive techniques.
>
> (1) In Table 1, MUSIC achieves better linear classification than Barlow Twins and VICReg.
>
> (2) In Table 2, MUSIC achieves the best results in most cases for transfer learning tasks (object detection, instance segmentation, and linear classification).
>
> (3) In Table 3, MUSIC achieves the best KNN classification on ImageNet among all methods including BYOL, SwAV, Barlow Twins, VICReg, etc., improving accuracy by 2+% over Barlow Twins and VICReg, which is a significant improvement.
>
> (4) In Table 5, MUSIC achieves superior results in most cases using different epochs compared with SimCLR, MoCo v2, BYOL, SwAV, and SimSiam.
>
> (5) In Table 7, MUSIC achieves consistently better results than VICReg using different feature dimensions.
>
> (6) In Table 10, MUSIC achieves better results in terms of both accuracy and efficiency than Barlow Twins and VICReg.
>
> **Q7: Regarding the even partition**
>
> A7: In this study, all segments have the same dimension, making the algorithm as simple as possible. They could have different dimensions given more application-specific prior. This issue can be further investigated.

---

> > ### Author Response · Authors · 2022-12-12
> > **Follow up before the discussion is closed**
> >
> > Thank you again for your feedback. As we are approaching the end of the discussion period, we would like to ask whether our revision and response have addressed your initial concerns about our work. We are more than happy to clarify and discuss more should you have additional questions.

---

### Official Review · Reviewer_yN39 · 2022-10-24

**Confidence:** 3
**Correctness:** 3
**Technical Novelty And Significance:** 3
**Empirical Novelty And Significance:** 3
**Recommendation:** 5

**Clarity, Quality, Novelty And Reproducibility:**

Clarity
- The paper is generally easy to follow but there are some uncertain part
- (section 1) The introduction section has too many references which takes the sentences apart.  In addition, the parenthesis for cited papers are missing for some reason.
- (section 2.2) "To make attributes discrete and complementary, one-hot encoding is applied to each segment". The one-hot encoding is implemented by the softmax in eq (1) as a "soft" one-hot encoding?  The pseudocode in appendix A also does not include any line of code for one-hot encoding.
- As stated above, it is not clear that the model learn disentangled visual attributes over the evenly divided segments. The visual analogy in Figure 1 can mislead what the model actually does.

Quality / Novelty
- The performance of the proposed method is comprehensively validated through the downstream tasks and empirical study
- The entropy loss was not very novel idea, as it has been used as a regularization term in a lot of previous work. However, the use was theoretically explained and the experimental result clearly show the effectiveness

Reproducibility
- The pseudocode helps understanding the algorithm but it would be even better to share the source code
- The implementation detail section has sufficient information for reproducing the model training

Minor comments
- Check the parenthesis for cited papers
- (page 9) "with inter-product" --> "with inner-product" ?


**Strength And Weaknesses:**

Strengths
- The information-theoretic entropy loss is well supported by the theoretical analysis
- Model training is efficient as it does not require a large match memory to include negative samples
-The overall performance is comparable to or outperforms previous states-of-the-arts in several downstream tasks
- The empirical study is comprehensive to understand the capability of the proposed method.

Weakness
- Some related work is missing. Recently, there have been a decent number of dense contrastive learning which divides the embedding space into small multi-vectors and exploits their correspondence. For, example,"Dense Contrastive Learning for Self-Supervised Visual Pre-Training", Xinlong Wang, Rufeng Zhang, Chunhua Shen, Tao Kong, Lei Li, CVPR, 2021,  "Dense Siamese Network for Dense Unsupervised Learning", Wenwei Zhang, Jiangmiao Pang, Kai Chen, Chen Change Loy, ECCV, 2022
- The paper argues that MUSIC divides an embedding vector into multiple segments that represent different types of attributes such as object part, texture, and shape. However, this hypothesis was never proved. There is no insightful analysis that the multiple segments capture the different visual attributes exclusively.  Although the hypothesis and analogy in Figure 1 may help delivering the "intended idea", it would be risky to insist that the model actually learn the features in that manner.
- Some part of writing is not clear (detailed below)



**Summary Of The Paper:**

This paper presents a self-supervise learning method using an entropy loss across multi-segments embedding vectors (MUSIC). This method automatically learn different types of attributes in each of multi segments.  Unlike the contrastive learning and its variants, it does not require negative samples in a large batch memory and asymmetric network, gradient stopping, or momentum update. Theoretical analysis support that the MUSIC embedding is transform-invariant, non-trivial, diverse, and discriminative. A set of downstream tasks show that the propose representation learning method is comparable to or outperforms previous states-of-the-arts. An empirical analysis investigates the efficiency in training , the effect of projector depth and feature dimension, and the necessity of the proposed loss function.




**Summary Of The Review:**

This paper seems to be a good contribution as a self-supervised learning method. However, the motivational idea, which the model learns disentangled attributes over segmented embedding space, was not validated. Also, some relevant references are missing. I hope the authors revise the current version to make up the weaknesses.

---

> ### Author Response · Authors · 2022-11-15
> **Response to Reviewer yN39**
>
> We would like to thank the reviewer $\textcolor{blue}{yN39}$ for the valuable feedback. Per your advice, we have provided more details and justifications for MUSIC.
>
> **Q1**: _Some related work is missing. Recently, there have been a decent number of dense contrastive learning which divides the embedding space into small multi-vectors and exploits their correspondence._
>
> **A1**: We have cited the papers and discussed the related work in Appendix D in the revision. The methods for dense prediction tasks spatially divide the image into sub-regions to enhance pixel- and region-level constraints. MUSIC is fundamentally different from these methods. MUSIC is to discretize each feature variable by a segment of one-hot vector, and the whole embedding space consists of multiple one-hot vectors representing various semantics. Synergistically, MUSIC can be used for dense prediction tasks by discretizing the feature vector of each sub-region and learning dense representations in the information-theoretic framework.
>
> **Q2**: _The paper argues that MUSIC divides an embedding vector into multiple segments that represent different types of attributes such as object part, texture, and shape. However, this hypothesis was never proved. There is no insightful analysis that the multiple segments capture the different visual attributes exclusively. Although the hypothesis and analogy in Figure 1 may help deliver the "intended idea", it would be risky to insist that the model actually learns the features in that manner._
>
> **A2**: Two generally desirable properties are illustrated in Fig. 1: 1) within each segment, samples can be classified into a set of different and discrete attributes; and 2) different segments represent different classification principles, which means that the mutual information between different segments is minimized, or equivalently the information/entropy is maximized. Both these two general properties are justified in our theoretical analysis. Specifically, we have demonstrated that the optimized embeddings with the entropy loss function are associated with the following characteristics: 1) each segment is one-hot encoded and samples are evenly assigned into all discrete units within each segment; and 2) the mutual information between different segments approaches zero.
>
> Per the reviewer’s advice, we have visualized representative samples in terms of the learned MUSIC embeddings in Appendix C, showing that meaningful features, such as different textures, shapes, and patterns, are indeed learned in the different segments as intended.
>
> **Q3**: _(section 1) The introduction section has too many references which takes the sentences apart. In addition, the parenthesis for cited papers is missing for some reason._
>
> **A3**: We have improved the sentences and fixed the issue with parentheses.
>
> **Q4**: _(section 2.2) "To make attributes discrete and complementary, one-hot encoding is applied to each segment". The one-hot encoding is implemented by the softmax in eq (1) as a "soft" one-hot encoding? The pseudocode in appendix A also does not include any line of code for one-hot encoding._
>
> **A4**: The one-hot encoding is implemented with softmax for each segment but the embedding vectors are optimized with our entropy-based loss for “hard” one-hot encoding as demonstrated in the theoretical analysis. For the pseudocode in Appendix A, the embedding vector $x \in R^{N\times D}$, N is the batch size, D is the total dimension, is first reshaped to multiple segments $x \in R^{N\times S\times D_s}$, S is the number of segments and $D_s$ is the dimension of each segment; i.e., `x=torch.reshape(x, [N, -1, D_s])`. Then, each and every segment is encoded with softmax, i.e., `q=torch.softmax(x, dim=2)`.
>
> **Q5**: _The entropy loss was not very novel idea, as it has been used as a regularization term in a lot of previous work. However, the use was theoretically explained and the experimental results clearly show the effectiveness._
>
> **A5**: Although the entropy loss has been widely used in other studies, MUSIC is the first scheme demonstrating that using the entropy loss only can learn meaningful representations in the self-supervised mode. This is because MUSIC discretizes the feature variables and makes the probability distribution estimable, and thus theoretical measures (entropy/mutual information/independence) can be directly used for both numerical optimization and theoretical analysis. Therefore, MUSIC presents a new embedding scheme and a new information-theoretic framework for self-supervised learning.
>
> **Q6**: _The pseudocode helps understand the algorithm but it would be even better to share the source code_
>
> **A6**: We have shared our source codes for training, testing, and visualization in the supplementary material.
>
> **Q7**: _(page 9) "with inter-product" --> "with inner-product" ?_
>
> **A7**: The typo has been corrected.

---

> > ### Author Response · Authors · 2022-12-12
> > **Follow up before the discussion is closed**
> >
> > Thank you again for your feedback. As we are approaching the end of the discussion period, we would like to ask whether our revision and response have addressed your initial concerns about our work. We are more than happy to clarify and discuss more should you have additional questions.

---

### Author Response · Authors · 2022-11-15
**General Responses**

$\newcommand{Ro}{\textcolor{orange}{ZcT3}}$ $\newcommand{Rg}{\textcolor{green}{f3VU}}$ $\newcommand{Rb}{\textcolor{blue}{yN39}}$
We would like to sincerely thank all reviewers for their insightful questions and constructive comments. We are happy to hear that the reviewers found MUSIC new and very interesting (Reviewer $\Ro$ and $\Rg$), effective (Reviewer $\Rb$, $\Ro$, and $\Rg$), theoretically grounded (Reviewer $\Rb$, $\Ro$, and $\Rg$), and has a wide scope (Reviewer $\Rg$), the empirical study is comprehensive and helpful (Reviewer $\Rb$, $\Ro$, and $\Rg$), and the overall performance is comparable to or better than the state-of-the-arts on comprehensive down-stream tasks (Reviewer $\Rb$).

We have learned one major concern raised by all reviewers, that is, “ _What embedding features are really learned? Are the learned features consistent with the illustration in Fig. 1?_ ” We have now addressed this concern from both theoretical and qualitative analysis.

**Theoretically**, we have demonstrated two general properties behind the illustration in Fig. 1 for the optimized MUSIC embeddings: 1) samples are discriminated with a set of discretized units in each segment, which corresponds to a set of discriminative attributes over each segment in Fig. 1; and 2) the mutual information between different segments is minimized so that multiple segments discriminate samples with different criteria, which correspond to different types of attributes in Fig. 1.

**Qualitatively**, as the reviewers suggested, we have visualized representative images assigned to some specific units in different segments in Figure C.4 in the revised Appendix, showing that meaningful features, such as different textures, shapes, and patterns, are indeed learned in different segments as intended.

Below we have responded to each and every critique from all reviewers and accordingly revised the manuscript by providing more details and justifications. The modified contents are highlighted in red in the revision and summarized as follows.

* Descriptions of MUSIC have been clarified and improved in the INTRODUCTION, ANALYSIS, and DISCUSSION and CONCLUSION sections.

* More qualitative visualization results have been added and discussed in Appendix C to justify that meaningful features are learned.

* Thorough discussion and comparison between MUSIC and related work have been added in Appendix D.

* More experimental results have been added and discussed in Appendix H.

* Typos have been corrected.

We have shared the source code of MUSIC for training, testing, and visualization in the **supplementary material**.

---

### Decision · Program_Chairs · 2023-01-20

**Decision:**

Reject

**Justification For Why Not Higher Score:**

The main claim of the paper is about the structure that the features contain. The proposed loss should encourage different attributes being encoded in different segments, however that property was only validated in a qualitative analysis. Without that, the proposed method is a structured variant of previous work, and does not offer significant improvements over previous methods.

**Justification For Why Not Lower Score:**

N/A

**Metareview: Summary, Strengths And Weaknesses:**

This paper proposes a method for SSL that allows to train multi-segment features. The criterion is setup so as different types of attributes are encoded in each of segments. This is obtained by maximising the entropy between segments. The reviewers have praised the proposed idea, and found it interesting and the experiments show that the trained models are non trivial. However, a concern that was raised by reviewers and is also shared by me, is that the main claim of the paper, namely the fact that attributes will be encoded in different segments is poorly tested. Indeed, the authors propose a qualitative analysis that hints at the fact that the relevant structure can be found in the representations, but not much more. The proposed method can be seen as a “structured” variant of VicREG or Barlow Twins from a high-level point of view; the structural contribution should have been tested in a much more thorough way. Given the feedback, rebuttal and discussion I recommend this paper for rejection, but encourage the authors to improve the manuscript and resubmit to another venue.